# Posterior parietal cortex estimates the relationship between object and body location during locomotion

Daniel S Marigold[1], Trevor Drew[2,3]*

[1]Department of Biomedical Physiology and Kinesiology, Simon Fraser University, British Columbia, Canada; [2]Département de Neurosciences, Université de Montréal, Québec, Canada; [3]Groupe de Recherche sur le Système Nerveux Central, Université de Montréal, Québec, Canada

**Abstract** We test the hypothesis that the posterior parietal cortex (PPC) contributes to the control of visually guided locomotor gait modifications by constructing an estimation of object location relative to body state, and in particular the changing gap between them. To test this hypothesis, we recorded neuronal activity from areas 5b and 7 of the PPC of cats walking on a treadmill and stepping over a moving obstacle whose speed of advance was varied (slowed or accelerated with respect to the speed of the cat). We found distinct populations of neurons in the PPC, primarily in area 5b, that signaled distance- or time-to-contact with the obstacle, regardless of which limb was the first to step over the obstacle. We propose that these cells are involved in a sensorimotor transformation whereby information on the location of an obstacle with respect to the body is used to initiate the gait modification.

DOI: https://doi.org/10.7554/eLife.28143.001

*For correspondence:
trevor.drew@umontreal.ca

**Competing interests:** The authors declare that no competing interests exist.

## Introduction

Navigation in cluttered environments dictates that animals and humans determine their relationship to stationary and moving objects for the purposes of avoidance or interception; these behaviors are essential for survival. Everyday examples of such activities range from the simple, such as stepping over a stationary obstacle and stepping up or down from a curb, to the more complex, such as adjusting one's gait to kick a moving soccer ball or stepping on or off a moving conveyor belt at the airport. Inherent in this process is the requirement to detect the presence of an obstacle, estimate its location with respect to the body, take into account the rate of gap closure (between body and object), and then use this information to appropriately modify the gait pattern. We suggest that the posterior parietal cortex (PPC) makes an essential contribution to this process.

Gibson, in his seminal work (*Gibson, 1958*), argued that animals could use optic flow to gauge distance and location to an obstacle, and thus modify gait to avoid it. Several studies have since confirmed this premise (*Prokop et al., 1997*; *Sun et al., 1992*; *Warren et al., 2001*). Later, *Lee (1976)* suggested that the brain extracts information about the time-to-contact (TTC) with an object from optic flow, a variable he called tau, and that this could be used to control gait. Again, multiple studies on locomotion (*Shankar and Ellard, 2000*; *Sun et al., 1992*) and other movements involving interception of moving targets (*Merchant et al., 2004*; *Merchant and Georgopoulos, 2006*) support this theory. However, it is important to note that tau is only one of several variables available to the brain to avoid an obstacle; distance and other time-related optical variables may also contribute (*Sun and Frost, 1998*; *Tresilian, 1999*).

In agreement with these behavioral studies, there is evidence from invertebrates to suggest that neurons can process optic flow for motor activities such as flight control, distance travelled, and

**eLife digest** Imagine crossing the street and having to step up onto a sidewalk, or running up to kick a moving soccer ball. How does the brain allow you to accomplish these deceptively simple tasks? You might say that you look at the target and then adjust where you place your feet in order to achieve your goal. That would be correct, but to make that adjustment you have to determine where you are with respect to the curb or the soccer ball. A key aspect of both of these activities is the ability to determine where your target is with respect to your current location, even if that target is moving. One way to do that is to determine the distance or the time required to reach that target. The brain can then use this information to adjust your foot placement and limb movement to fulfill your goal.

Despite the fact that we constantly use vision to examine our environment as we walk, we have little understanding as to how the brain uses vision to plan where to step next. Marigold and Drew have now determined whether one specific part of the brain called the posterior parietal cortex, which is known to be involved in integrating vision and movement, is involved in this planning. Specifically, can it estimate the relative location of a moving object with respect to the body?

Marigold and Drew recorded from neurons in the posterior parietal cortex of cats while they walked on a treadmill and stepped over an obstacle that moved towards them. On some tests, the obstacle was either slowed or accelerated quickly as it approached the cat. Regardless of these manipulations, some neurons always became active when the obstacle was at a specific distance from the cat. By contrast, other neurons always became active at a specific time before the cat met the obstacle. Animals use this information to adjust their gait to step over an obstacle without hitting it.

Overall, the results presented by Marigold and Drew provide new insights into how animals use vision to modify their stepping pattern. This information could potentially be used to devise rehabilitation techniques, perhaps using virtual reality, to aid patients with damage to the posterior parietal cortex. Equally, the results from this research could help to design brain-controlled devices that help patients to walk – or even intelligent walking robots.

DOI: https://doi.org/10.7554/eLife.28143.002

landing (**Baird et al., 2013**; **Egelhaaf and Kern, 2002**; **Fotowat and Gabbiani, 2011**; **Srinivasan and Zhang, 2004**). Similarly, the pigeon has neurons in the nucleus rotundus that can extract optic flow and other variables, such as tau, from visual stimuli and which could be used for object avoidance (**Sun and Frost, 1998**; **Wang and Frost, 1992**). In mammals, multiple cortical structures analyze optic flow, including the middle temporal (MT/V5) and medial superior temporal (MST) cortices (**Duffy and Wurtz, 1995**, **1997**; **Orban, 2008**), as well as the PPC, the premotor cortex, and even the motor cortex (**Merchant et al., 2001**, **2003**; **Schaafsma and Duysens, 1996**; **Siegel and Read, 1997**). However, the important question of how the mammalian nervous system uses optic flow information to guide movement has been studied in only a few behaviors (**Merchant and Georgopoulos, 2006**), and then only for arm movements. In this manuscript, we extend these studies by determining how neural structures treat visual information for the control of gait.

Our previous work demonstrates the presence of neuronal activity in the PPC that begins several steps before the step over the obstacle and that could contribute to planning (**Andujar et al., 2010**; **Drew and Marigold, 2015**; **Marigold et al., 2011**). In this manuscript, we test the hypothesis that the PPC contributes to obstacle avoidance by constructing an estimation of an approaching object's location relative to the body's current state, and in particular the diminishing gap between them and its relation to the ongoing step cycle of each limb (gap closure). We show the presence of neurons in the PPC that code either distance-to-contact (DTC) or TTC, and we suggest that this discharge represents the starting point of a complex sensorimotor transformation involved in planning the gait modification.

## Results

### Modifying obstacle speed dissociates DTC and TTC

We trained cats to step over moving obstacles attached to a treadmill. The cats performed the task in a relatively stereotypical manner (*Lajoie and Drew, 2007*). Measurements of the DTC and of the TTC with the obstacle both decreased monotonically to a value of ~0 at the onset of the step over the obstacle (*Figure 1A*).

To dissociate between cells potentially related to either DTC or TTC, we recorded cell activity in two complementary locomotor tasks. In one, the obstacle advanced toward the cat at the same speed as the treadmill belt on which the cat walked (matched task) while in the other, the obstacle advanced at a slower speed (visual dissociation task: *Drew et al., 2008*; *Lajoie and Drew, 2007*). As the speed of the treadmill on which the cat walked was the same in both tasks (0.45 m.s$^{-1}$ in these experiments), DTC and TTC are a function of the speed of the advancing obstacle (*Figure 1B*). For example, in the matched task, when DTC = 45 cm, TTC = 1000 ms. However, in the visual dissociation task (obstacle speed slowed to 0.3 m.s$^{-1}$), when DTC = 45 ms, TTC = 1500 ms.

This dissociation of DTC and TTC is a fundamental part of our analysis of the contribution of cells in the PPC to the estimation of gap closure. In brief, a cell in which the onset of activity is determined by TTC will discharge at the same time relative to the onset of the step over the obstacle in both the matched and the visual dissociation tasks (red vertical line at 1000 ms in *Figure 1C,D*). In contrast, during the visual dissociation task, a cell related to DTC would discharge relatively earlier, indicative of the longer time required to cover the fixed distance (green vertical line at 45 cm in *Figure 1C,D*).

### Neuronal database

The present report is based on 67 cells recorded from two cats (37 from cat PCM7 and 30 from PCM9), selected from a much larger database on the basis of the criteria provided in the Materials and methods. These neurons were primarily recorded from the posterior bank of the ansate sulcus, the adjoining lateral bank of the lateral sulcus and the adjacent gyrus between these two sulci, corresponding to area 5b of the PPC; some cells were also recorded from the border region of area 5/7 (see *Figure 2*). Some of these cells (42/67) were included in a previous report (*Marigold and Drew, 2011*). Note that we recorded all cells from the right PPC and that the left limb is therefore contralateral to the recording site.

### Cell discharge during matched and visual dissociation tasks

As indicated in the preceding section, a cell in which the change in discharge activity is related to the distance between the obstacle and the cat would be expected to become active earlier in the visual dissociation task than in the matched task. An example of such a cell is illustrated in *Figure 3*. In this example, cell discharge was low and tonic during unobstructed locomotion (blue traces) and then showed a distinct increase in discharge in the two steps before the obstacle both in the left limb leads condition and in the right limb leads condition. This discharge peaked just prior to the onset of the flexor muscle activity during the step over the obstacle (represented by the black vertical line) before decreasing to, or below, control levels (blue trace), as the lead limb stepped over the obstacle. This general behavior occurred in both the matched task (red traces) and in the visual dissociation task (green traces). The increase in cell discharge, as calculated from the average of the onsets in the individual trials (see *Figure 3—figure supplement 1*), began 514 ± 124 ms before the step over the obstacle in the left lead condition of the matched task and 511 ± 231 ms in the right lead condition of the same task (red vertical lines). In the visual dissociation task, the respective values were 745 ± 161 ms and 723 ± 187 ms (green vertical lines). A t-test showed a significant difference in the time of the onset of cell discharge between the matched and visual dissociation task, both for the left limb leads condition (p<0.001) and for the right limb leads condition (p=0.002).

The activity pattern illustrated in *Figure 3* is compatible with the hypothesis that DTC determined the onset of the discharge activity (see *Figure 1B–D*). Importantly, we also verified that this cell discharge was not step-related. As shown in *Figure 3D* (left), there is no relationship between the onset of cell discharge and the onset of the activity in the coBr muscle in the step before the obstacle (a and c in *Figure 3C*). Rather, as expected on the basis of the similarity in the time of onset of cell

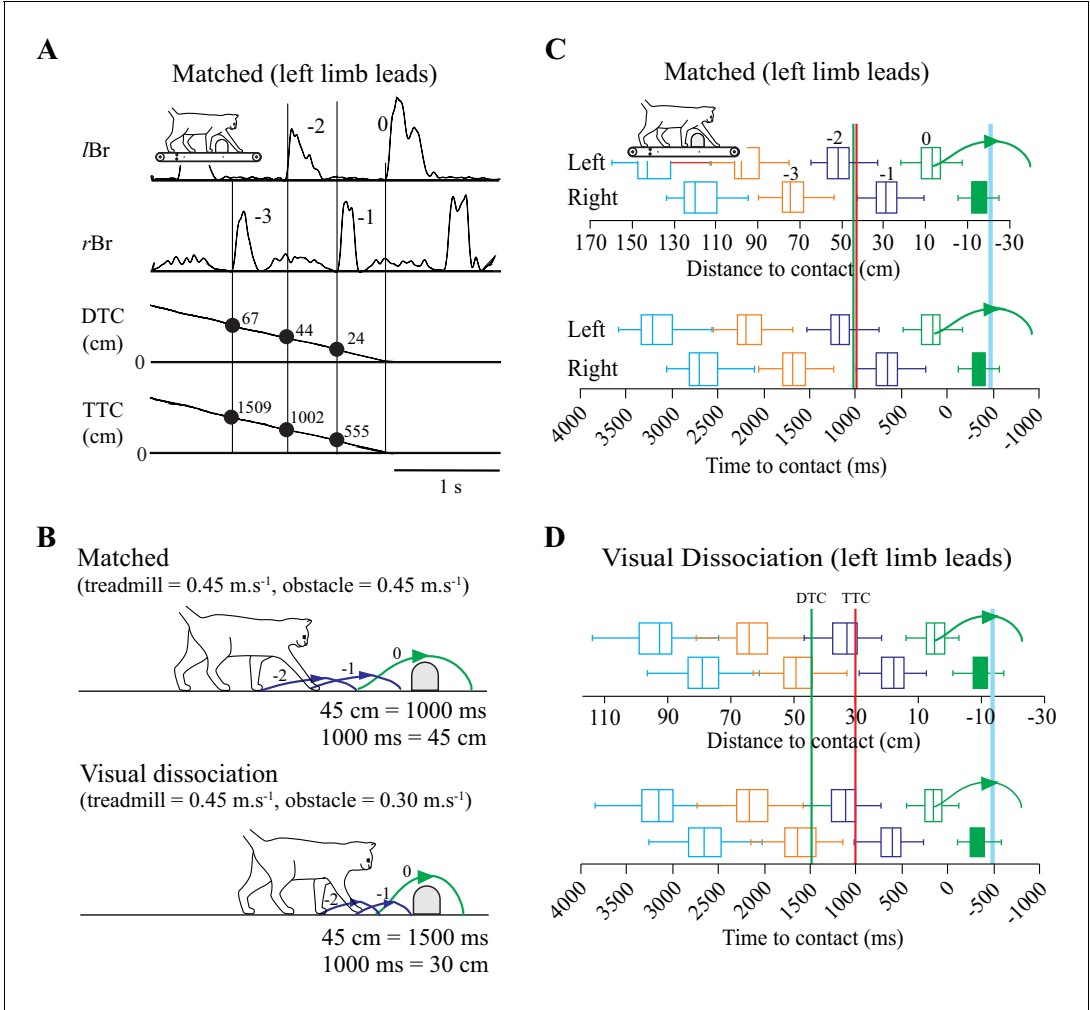

**Figure 1.** Behavioral Measures. (**A**) Section of data illustrating, from top to bottom: the activity of the left and right brachialis (*l*Br and *r*Br); the distance-to-contact (DTC) and the time-to-contact (TTC) with respect to the obstacle. Solid vertical lines are aligned with the onset of each Br and the numerical values indicate measures of DTC and TTC for three steps before the step over the obstacle. (**B**) schematic representation of the relationship between DTC and TTC during the matched and visual dissociation tasks (see text). Steps −2 to 0 correspond to those in A. C,D: Box plots indicating the DTC (top panel) and the TTC (bottom panel) as measured at the onset of the *l*Br ('left') and *r*Br ('right') in the steps preceding the step over the obstacle for each limb during the matched (**C**) and visual dissociation (**D**) tasks. Pairs of steps (left/right) are shown in the same color. Steps −3 to 0 in the top row of C correspond to those in A. Boxes include 50% of the values and the vertical line inside the box indicates the median of the values. Horizontal lines (whiskers) enclose 1.5 * interquartile range. Values greater than 1.5 * interquartile range have been removed (see Materials and methods). The horizontal scale for time is kept constant in C and D, and therefore, the scale for distance is expanded in D because the obstacle moves relatively more slowly in the visual dissociation task. Vertical green and red lines (**C,D**) indicate the theoretical onset of a cell related to either DTC or TTC, respectively (see text). The curved (green) line indicates the limb stepping over the obstacle (blue vertical line). C and D compiled from 127 and 167 trials, respectively, taken from 14 experimental sessions in cat PCM7. (***Figure 1—source data 1***).

DOI: https://doi.org/10.7554/eLife.28143.003

The following source data and soruce codes are available for figure 1:

**Source data 1.** Source data for box plots in ***Figure 1C,D***.
DOI: https://doi.org/10.7554/eLife.28143.004
**Source code 1.** Script for data in ***Figure 1***.
DOI: https://doi.org/10.7554/eLife.28143.005

discharge independent of which limb leads, the relationship to coBr forms two populations, separated by ~500 ms (~1 step or half the duration of the step cycle). Conversely, discharge onset in the cell overlaps with the onset of activity in the coBr when the left limb leads but with the iBr when the right limb leads (***Figure 3D***, *middle*). Overall, we found no constant relationship between the onset

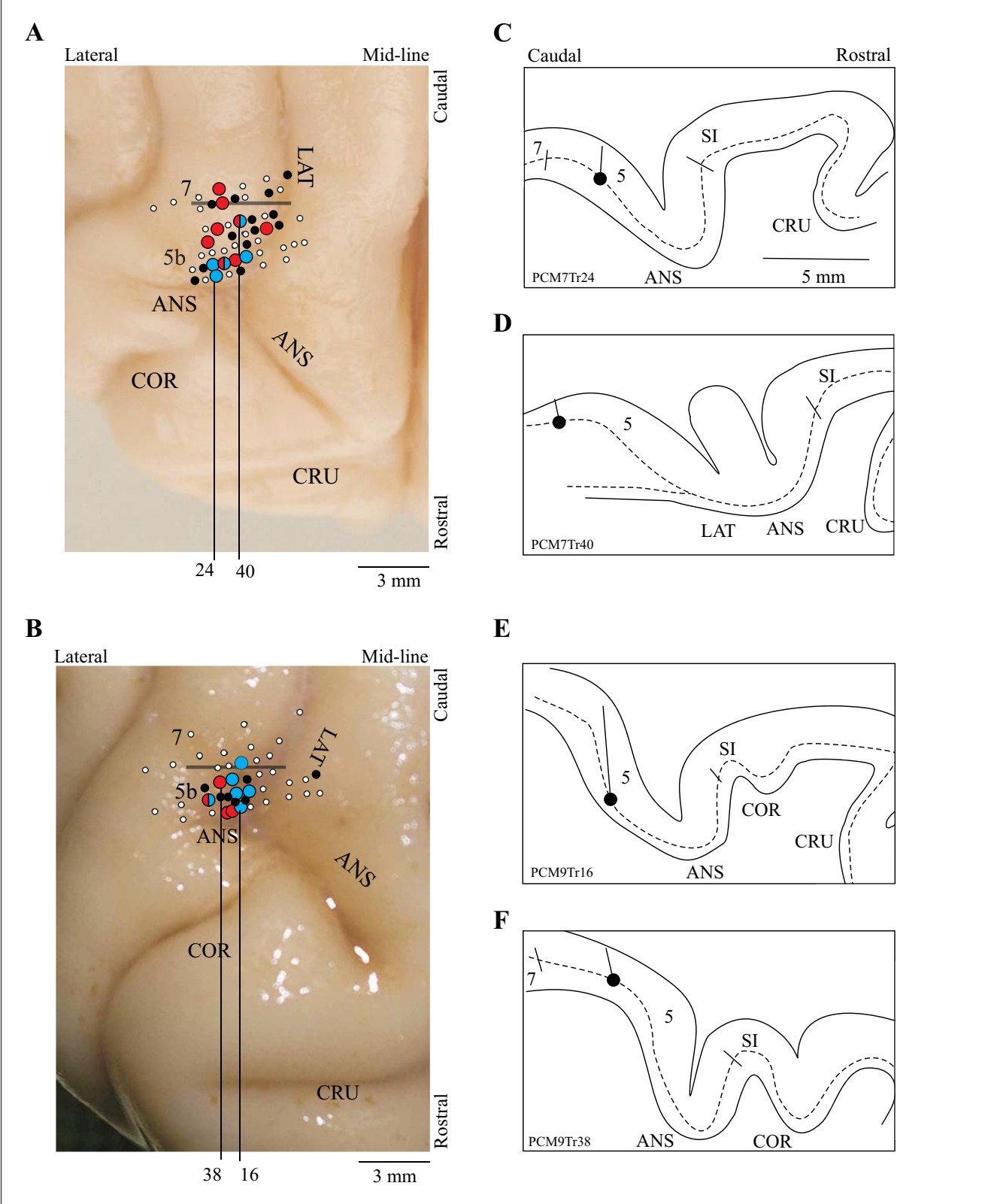

**Figure 2.** Location of recording tracks. (A,B) Postmortem photographs of the rostral aspect of the brains of PCM7 (A) and PCM9 (B) (dorsal views). Circles superimposed on these photographs show the location of the recording tracks in each animal: filled white circles = penetrations; filled black circles = location of step-advanced cells; blue circles = DTC-related cells; red circles = TTC-related cells; mixed red and blue circles = both DTC- and TTC-related cells recorded in the penetration. Horizontal bar indicates the approximate border between areas 5b and 7. (C-F) Tracings of the

*Figure 2 continued on next page*

*Figure 2 continued*

reconstructions of four penetrations (**C,D** from PCM7 and **E,F** from PCM9). The dotted line indicates the location of layer V and the filled circle indicates the recording site in the illustrated track. The medio-lateral location of each reconstruction is indicated by the black vertical lines on A,B; these lines originate from the penetration illustrated in each panel in C-F. Abbreviations: 5b, area 5b of the PPC; 7, area 7 of the PPC; ANS, ansate sulcus; LAT, lateral sulcus, COR, coronal sulcus; CRU, cruciate sulcus; SI, somatosensory cortex.
DOI: https://doi.org/10.7554/eLife.28143.006

of discharge activity and step-related activity in a given limb in this cell. In contrast, we found a significant relationship between the time of the end of cell discharge and the onset of activity in the flexor muscle of the lead limb during the step over the obstacle (*Figure 3D right*) as would be expected for a limb-independent cell (see Materials and methods). We observed similar relationships to those illustrated in *Figure 3D* in all cells in our database.

## Distinct populations of cells are related to DTC and TTC

The cell illustrated in *Figure 3*, putatively identified as related to distance based on its relatively earlier discharge onset in the visual dissociation task, is displayed in *Figure 4A* together with the averaged calculated traces of DTC and TTC. The vertical green and red lines intersect these traces at the average time of onset of the cell discharge as calculated from the individual trials (see *Figure 3*). Because the obstacle advanced more slowly in the visual dissociation task, the slope of the DTC trace in this task (diagonal green trace) is less than that in the matched task (diagonal red trace). As a consequence, cell discharge begins at approximately the same distance in the two tasks (horizontal black line). In contrast, the interception with the bottom pair of traces shows that cell onset begins at different TTCs in the two tasks. We quantified this relationship with a one-way ANOVA across the four situations, which demonstrated a significant effect for TTC (*Figure 4B* bottom). Post-hoc t-tests with Bonferroni corrections showed significant differences with task (matched versus visual dissociation) but no significant differences with condition (left or right lead). In contrast, the ANOVA showed a non-significant effect with DTC (*Figure 4B*, top), effectively indicating that the onset of cell discharge occurred when the obstacle was a constant distance from the cat, regardless of task or lead limb.

The opposite situation is shown for the cell in *Figure 4C*. In this case, we found a significant effect of DTC (*Figure 4D*, top) on the onset of cell discharge. Post hoc t-tests showed significant differences (asterisks) between the matched right lead situation and both conditions in the visual dissociation task. However, we found no significant effect of TTC on cell discharge (*Figure 4D*, bottom), indicating that the onset of cell discharge occurred when the obstacle was a constant TTC regardless of task or condition.

Overall, we identified two populations of cells related to gap closure: one group in which the onset of discharge activity occurred at a constant DTC, and a second group in which the onset of discharge occurred at a constant TTC. *Figure 5* summarizes the results of the ANOVAs calculated from a population of 51/67 cells that we recorded in all four situations (matched and visual dissociation task, left and right lead) and for which we could measure the onset of the period of discharge activity from individual trials during the step over the obstacle (see *Figure 3—figure supplement 1*). To identify DTC-related cells, we determined those cells—similar to the example in *Figure 4A*—in which cell discharge varied significantly with respect to TTC ($p<0.01$) but non-significantly with respect to DTC ($p>0.01$). A total of 14/51 cells fulfilled this criterion. These cells are plotted as the cyan traces in *Figure 5A* and are encompassed by the cyan square in *Figure 5C*. A further 15/51 cells fulfilled the inverse criterion (red traces in *Figure 5B* and red square in *Figure 5C*) and are identified as TTC-related cells. A further 6/51 cells showed significant effects ($p<0.01$) of both distance and time (green lines in *Figure 5A,B* left plots and green square in *Figure 5C*) and 16/51 cells showed non-significant effects of both time and distance (gray lines in *Figure 5A,B*, middle and right, and gray square in *Figure 5C*).

As indicated in *Figure 2*, cells defined as DTC- and TTC-related were found throughout the explored region of area 5b of the PPC as well as in the border area between areas 5 and 7. No clustering of categories was observed. Cells of both categories were recorded from each cat (9 TTC-

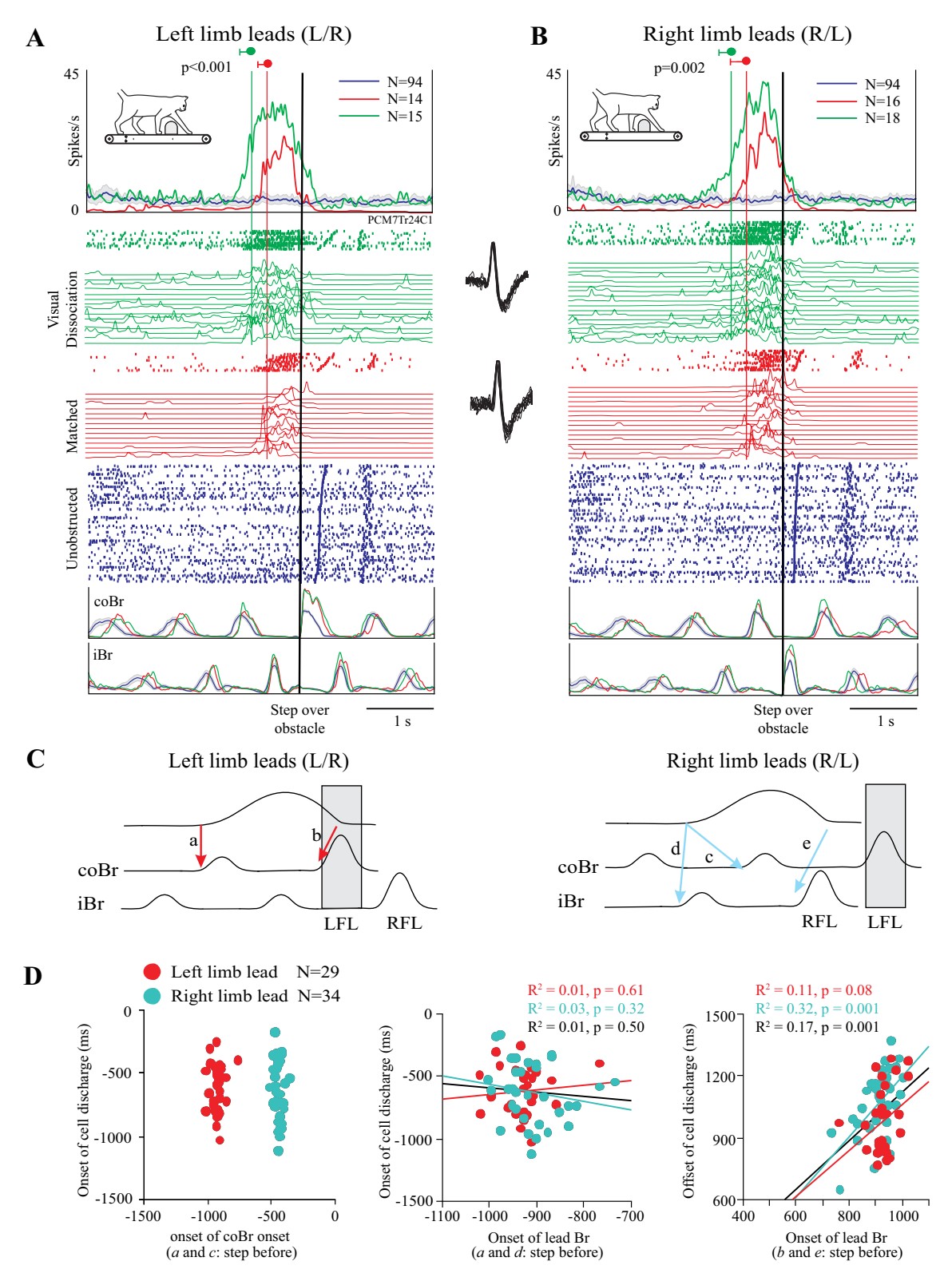

**Figure 3.** Discharge of an example cell during the matched and visual dissociation tasks. (A,B) Cell discharge during left and right limb lead condition. For each task, we illustrate the average discharge in the form of peri-event histograms (PEHs), together with the raster of cell activity, the instantaneous frequency during each trial, and the averaged activity of the coBr and iBr for the matched task (red), visual dissociation task (green), and for unobstructed walking (blue). Data are synchronized to the onset of activity (vertical black line) in the coBr (A) or the iBr (B) and displayed for 3200 ms

*Figure 3 continued on next page*

*Figure 3 continued*

prior to EMG onset and 2000 ms after. Vertical green and red lines and values indicate the average onset of cell activity during the visual dissociation and matched task, respectively, as calculated from the onset of the cell discharge in individual trials (see *Figure 3—figure supplement 1*). Insets between A and B show cell waveform during recordings of the matched (bottom trace) and visual dissociation (top trace) trials. (C) Schematic illustration showing selected temporal relationships tested between cell and muscle activity (see D). Shaded rectangle indicates the step over the obstacle by the left forelimb (LFL). RFL = right forelimb. (D) Linear regressions for cell onset vs. onset of the coBr during the step over the obstacle for the left and right limb lead condition (*left*); the relationship of cell onset to the onset of the flexor EMG in the lead limb in the step before the step over the obstacle (*middle*); and the end of cell discharge as a function of the onset of the activity in the Br of the lead limb during the step over the obstacle (*right*). Note that in the latter graph, measures are relative to the onset of activity in the Br in the preceding step. Data and linear regressions are shown for the left (red) and the right (cyan) limb lead conditions, together with the combined linear regression (black). (*Figure 3—source data 1*). *Figure 3—figure supplement 1*: Detection of bursts of unit activity in individual trials.

DOI: https://doi.org/10.7554/eLife.28143.007

The following source data, source code and figure supplements are available for figure 3:

**Source data 1.** Source data for graphs in *Figure 3D*.
DOI: https://doi.org/10.7554/eLife.28143.009
**Source code 1.** Script for data in *Figure 3*.
DOI: https://doi.org/10.7554/eLife.28143.010
**Figure supplement 1.** Detection of bursts of unit activity in individual trials.
DOI: https://doi.org/10.7554/eLife.28143.008

related cells from cat PCM7 and 6 from cat PCM9; 6 DTC-related cells from cat PCM7 and 8 from cat PCM9: see *Figure 2* for distribution).

To determine the extent to which these two populations are distinct, we calculated an index (see Materials and methods), based on the difference in standardized discharge rate between the visual dissociation and the matched task, for all cells showing a significant relationship to TTC or DTC. As illustrated in *Figure 5D*, the two populations (with the exception of 1 cell) were clearly separated one from the other, reflecting the difference in their discharge profiles with respect to DTC and TTC. Cells showing a significant relationship with both TTC and DTC (green circles) divided into one group or the other. A bootstrapping exercise performed on the DTC- and TTC-related cells supports our contention that such cells form two distinct categories (*Figure 5—figure supplement 1*). Of the cells showing a constant relationship to TTC across conditions and task, most began to discharge when the obstacle was between 300 and 1000 ms from the cat (*Figure 5B*, *right* and 5E, *top*). The cells related to DTC discharged when the obstacle was 20 to 40 cm away from the cat (*Figure 5A*, *right* and *Figure 5E*, *bottom*). These values are compatible with cell discharge beginning one or two steps before the step over the obstacle.

The fact that cell onset occurred at varying DTC and TTC (*Figure 5E*) implies that there is a sequential activation of cells included within each of our two major populations. This is illustrated in *Figure 6A,D* for the matched task for five representative DTC-related cells (A) and five TTC-related cells (D). When considering the overall discharge activity of both the population of DTC-related cells (*Figure 6B,C*) and of the TTC-related cells (*Figure 6E,F*), this staggered onset, together with the progressive increase in discharge observed within individual cells (*Figures 3* and *4*), resulted in a prolonged and progressive increase in discharge activity, for both left and right limb lead conditions. This ramp increase begins two to three steps before the step over the obstacle for the DTC-related population and slightly later for the TTC-related population. Moreover, as expected on the basis of the individual examples, the onset of this increase in activity occurs earlier in the visual dissociation task (green traces) than in the matched task (red traces) for the DTC-related cells, but at the same time for the population of TTC-related cells. The peak of the discharge activity, for both populations, occurs just before or after the onset of the gait modification. It is also noticeable in *Figure 6B,C* that the onset of the change in activity in the DTC-related population activity in the visual dissociation condition actually begins five to six steps before the step over the obstacle, well before the more prominent burst of activity on which we concentrate. This early increase in discharge is not simply an effect of smearing but represents a propensity for some of this population of cells to show a more tonic increase in the visual dissociation task (see *Figure 6—figure supplement 1*).

Using all the cells included in the analysis for *Figure 5* in the population averages did not change the general form of the discharge (see *Figure 6—figure supplement 1*). The populations still

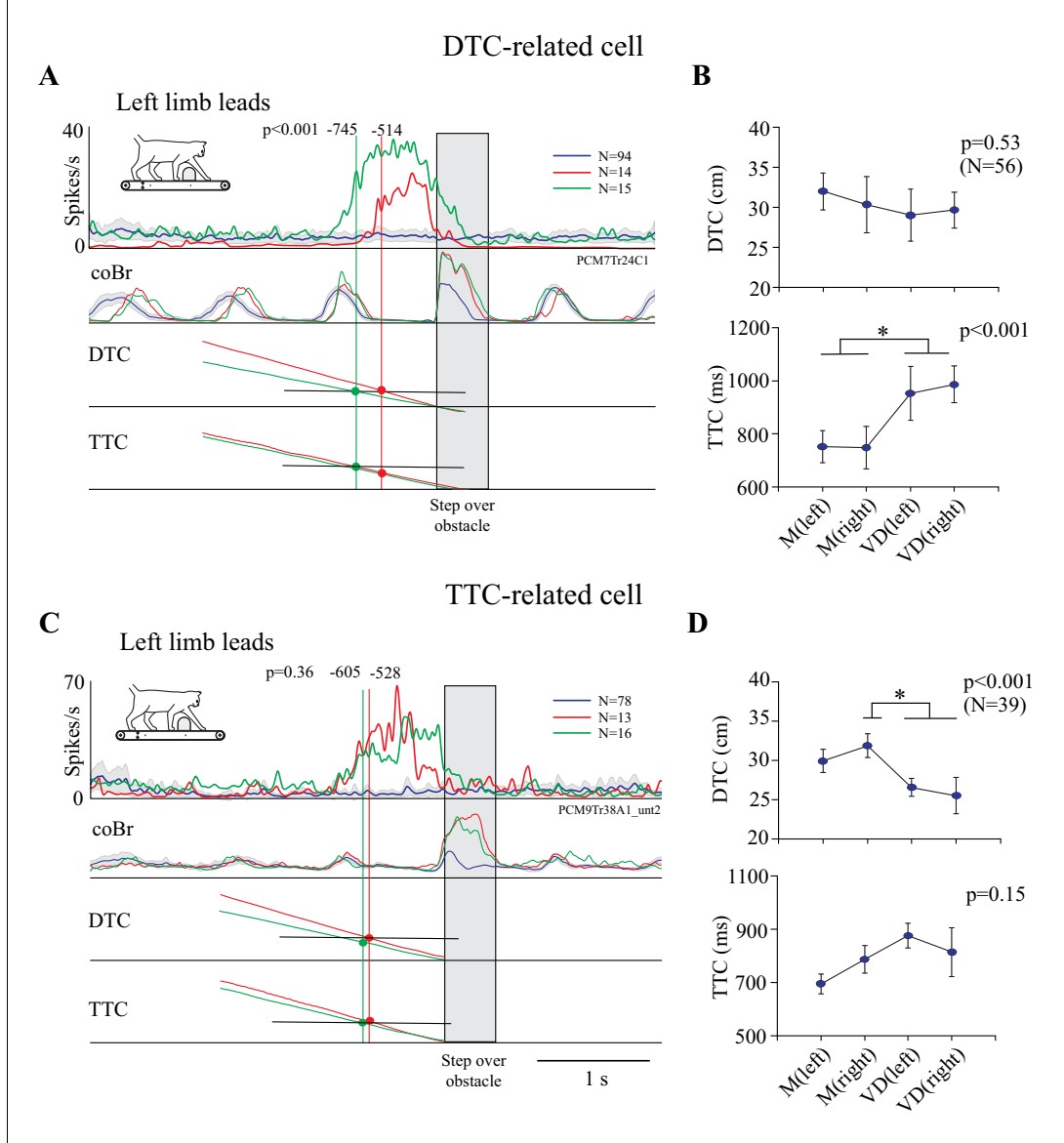

**Figure 4.** Relationship of cell discharge onset to distance and time (individual examples). **A and C** show two examples of cell discharge in the unobstructed (blue line), matched (red), and visual dissociation (green) task in the left limb leads condition, together with traces indicating DTC and TTC of the head from the obstacle. A is the same cell as in *Figure 3*. The vertical green and red lines are aligned with the average onset of cell discharge as calculated from individual trials (values indicated at the top of each line). Their intersection with the DTC and TTC traces is indicated with colored circles. The probability that the onset of cell activity during the matched and visual dissociation tasks is the same is indicated by the p-value. (B, D) Plots of the average DTC (top graph) and TTC (bottom graph) (±interval of confidence at p=0.05) for the matched and visual dissociation task in the lead and trail conditions. Results of an ANOVA are shown to the top right of each graph and asterisks indicate significant differences between left and right lead and matched (M) and visual dissociation (VD) tasks (Bonferroni correction, p<0.05). Plots for DTC and TTC in B,D are scaled to the same range (*Figure 4—source data 1* and *2*).

DOI: https://doi.org/10.7554/eLife.28143.011

The following source data and soruce codes are available for figure 4:

**Source data 1.** Source data for *Figure 4B*.
DOI: https://doi.org/10.7554/eLife.28143.012
**Source data 2.** Source data for *Figure 4D*.
DOI: https://doi.org/10.7554/eLife.28143.013
**Source code 1.** Script for data in *Figure 4*.
DOI: https://doi.org/10.7554/eLife.28143.014

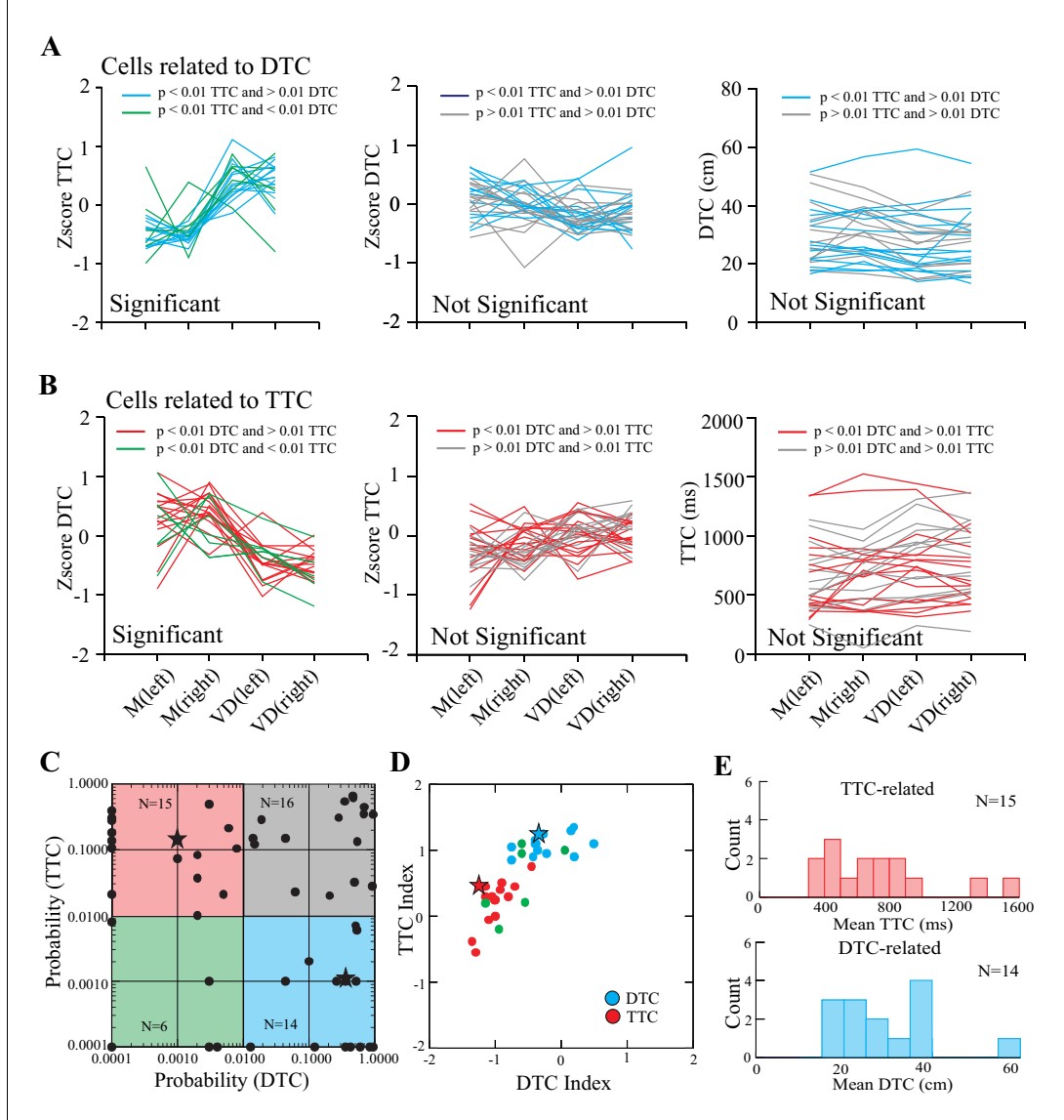

**Figure 5.** Relationship of cell discharge onset to distance and time (population analysis). (**A**) Cells showing both a significant effect of TTC (*left*, cyan lines) and a non-significant effect of DTC (*middle and right*, cyan lines). (**B**) Cells showing a significant effect of DTC (*left*, red lines) and a non-significant effect of TTC (*middle and right*, red lines). Green lines in A, B (*left*) indicate cells showing a significant relationship to both DTC and TTC. Gray lines (*middle and right*) indicate cells showing a non-significant relationship with each. Magnitude of the response is plotted as a Z score in the left and middle plots and as absolute values on the right. (**C**) Probability of a significant relationship with TTC as a function of the probability of a significant relationship to DTC (log scales). Cyan rectangle illustrates the 14 DTC-related cells and the red rectangle illustrates the 15 TTC-related cells. Cells that had no relationship to either are clustered in the top right, whereas those with a significant relationship to both are in the bottom left. (**D**) Modulation index (see Materials and methods) for the DTC and TTC cells; green symbols indicate cells with a significant relationship to both. Asterisks in C and D indicate the two cells illustrated in *Figure 4*. (**E**) Histograms illustrating the distribution of values for TTC- and DTC-related cells. (*Figure 5—source data 1* and *2*). *Figure 5—figure supplement 1*: Bootstrapped data for index of TTC- and DTC-related cells.

DOI: https://doi.org/10.7554/eLife.28143.015

The following source data, source code and figure supplements are available for figure 5:

**Source data 1.** Source data for graph in *Figure 5D*.
DOI: https://doi.org/10.7554/eLife.28143.017
**Source data 2.** Source data for graphs in *Figure 5C,E*.
DOI: https://doi.org/10.7554/eLife.28143.018
**Source code 1.** Script for data in *Figure 5*.
DOI: https://doi.org/10.7554/eLife.28143.019
**Figure supplement 1.** Bootstrapped data for index of TTC- and DTC-related cells.

*Figure 5 continued on next page*

*Figure 5 continued*

DOI: https://doi.org/10.7554/eLife.28143.016

showed a clear ramp discharge that peaked at around the time of the onset of the gait modification. This suggests that even those cells without a significant relationship to DTC or TTC participate in the planning of the gait modification.

## The relationship to DTC and TTC is maintained during obstacle acceleration

To further probe the relationship between cell activity and gap closure, we also manipulated the relationship between DTC and TTC by accelerating the obstacle several steps prior to the step over it. This acceleration, which we always applied during the visual dissociation task, produced major changes in the organization of the sequence of steps prior to the step over the obstacle, as illustrated in *Figure 7A,B*. In particular, in all cases, the acceleration produced a change in the sequence of steps such that the limb that stepped over the obstacle was the opposite of that predicted on the basis of the unperturbed sequence.

As an example, in *Figure 7A*, the top illustration represents the step sequence during the unperturbed situation in the visual dissociation task (right limb lead). The sequence of steps is regular, and the cat places the left paw just in front of the obstacle before stepping over it with the right forelimb (green curved arrow). The next sequence down (condition 1L) shows the situation when we applied the obstacle acceleration at the onset of the left stance of the left limb, three steps before the predicted step over the obstacle, as indicated by the filled orange box. This accelerated the obstacle quickly toward the cat so that instead of lifting up the left limb in step −1 and placing it in front of the obstacle, as in the unperturbed situation, it instead stepped over the obstacle (see also *Figure 7—figure supplement 1*). In the third sequence (condition 2L), we initiated the acceleration two steps earlier (−5, filled cyan box). As in the preceding sequence, the acceleration of the obstacle reduced the distance between the cat and the obstacle and reset the step sequence, again resulting in the cat stepping over the obstacle with the left limb. Note that the distance of the obstacle from the cat in the right limb in step −4 is similar in all three sequences (vertical orange line) while in step −1 the sequence is reversed with respect to that seen in the unperturbed situation (green vertical line), supporting the assertion that the acceleration reversed the sequence of steps over the obstacle.

We observed a complementary situation for step sequences in which the cat would normally step over the obstacle with the left limb in the absence of the acceleration (*Figure 7B*). For example, in the second trace down (condition 1R), the sudden acceleration of the obstacle in step −2 resulted in the cat lifting the right limb (step −1) over the obstacle instead of placing it down and taking an extra step as it did in the unperturbed situation. In the 2R condition, an acceleration applied in step −4 (filled orange box) likewise resulted in the loss of a step and a reversal of the expected pattern of activity. As in *Figure 7A*, the orange and green vertical lines demonstrate the reversal of the sequence.

One of the major effects of the acceleration was to decrease the time taken to close the gap between the cat and the obstacle for a given DTC or TTC. In the example illustrated in *Figure 7C* (same DTC-related cell as in *Figure 3*), cell discharge in the unperturbed visual dissociation task (green trace) began 745 ms before the step over the obstacle and at a DTC of 30.6 cm. In the acceleration task, we applied the acceleration in step −3 of the coBr (orange box), when the obstacle was 39.4 cm from the cat. As the obstacle accelerated toward the cat, the cell started to discharge at a DTC of 30.2 cm. However, because of the acceleration, this discharge occurred only 403 ms before the cat stepped over the obstacle, resulting in the relative delay of the onset of cell discharge in the acceleration task (purple trace) compared to the visual dissociation task (see *Figure 7—figure supplement 2*). However, the projection of the average cell onset in the three tasks (vertical lines) onto the DTC traces confirms that the discharge in all three tasks occurred at the same DTC (see values at top left of *Figure 7C*: DTC). In contrast, the projection onto the TTC trace shows that cell onset varied between the three tasks.

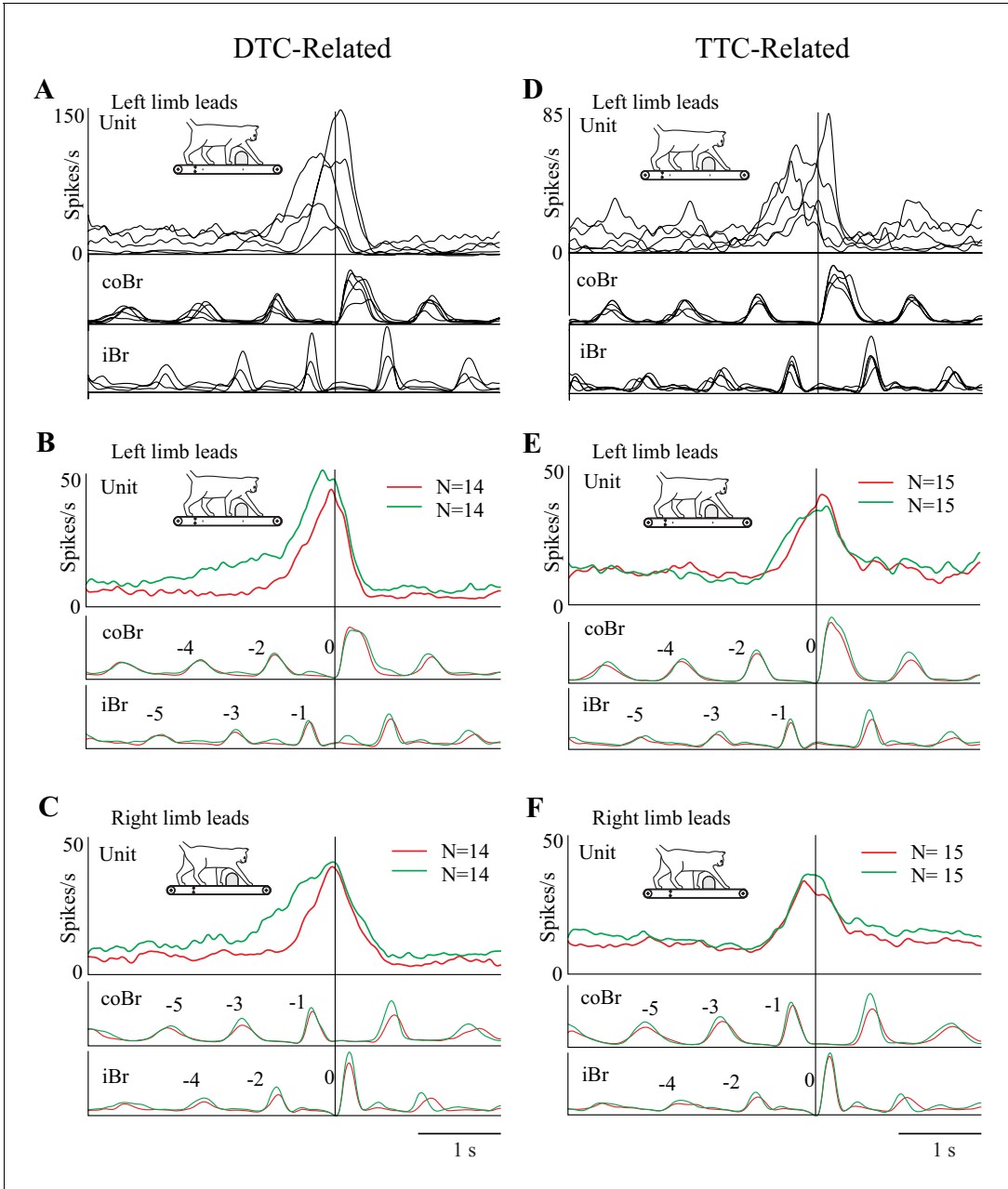

**Figure 6.** Relationship of cell discharge onset to distance and time (population averages). (A,D) Five example cells illustrating that the changes in cell discharge in the different populations begin at staggered times preceding the step over the obstacle. Cell discharge patterns taken from the matched condition and aligned to the onset of the Br. Cells scaled to the cell with the highest discharge rate in each illustrated group of cells. (B,C) Average discharge activity of the 14 DTC cells during left (B) and right (C) lead conditions. (E,F) Average discharge activity of the 15 TTC cells during left (E) and right (F) lead conditions. Data in B,C,E,F are shown for the matched (red traces) and visual dissociation (green traces) task. All traces are scaled identically in (B,C) and (E,F). *Figure 6—figure supplement 1*: Additional Population Averages.

DOI: https://doi.org/10.7554/eLife.28143.020

The following figure supplement is available for figure 6:

**Figure supplement 1.** Additional population averages.

DOI: https://doi.org/10.7554/eLife.28143.021

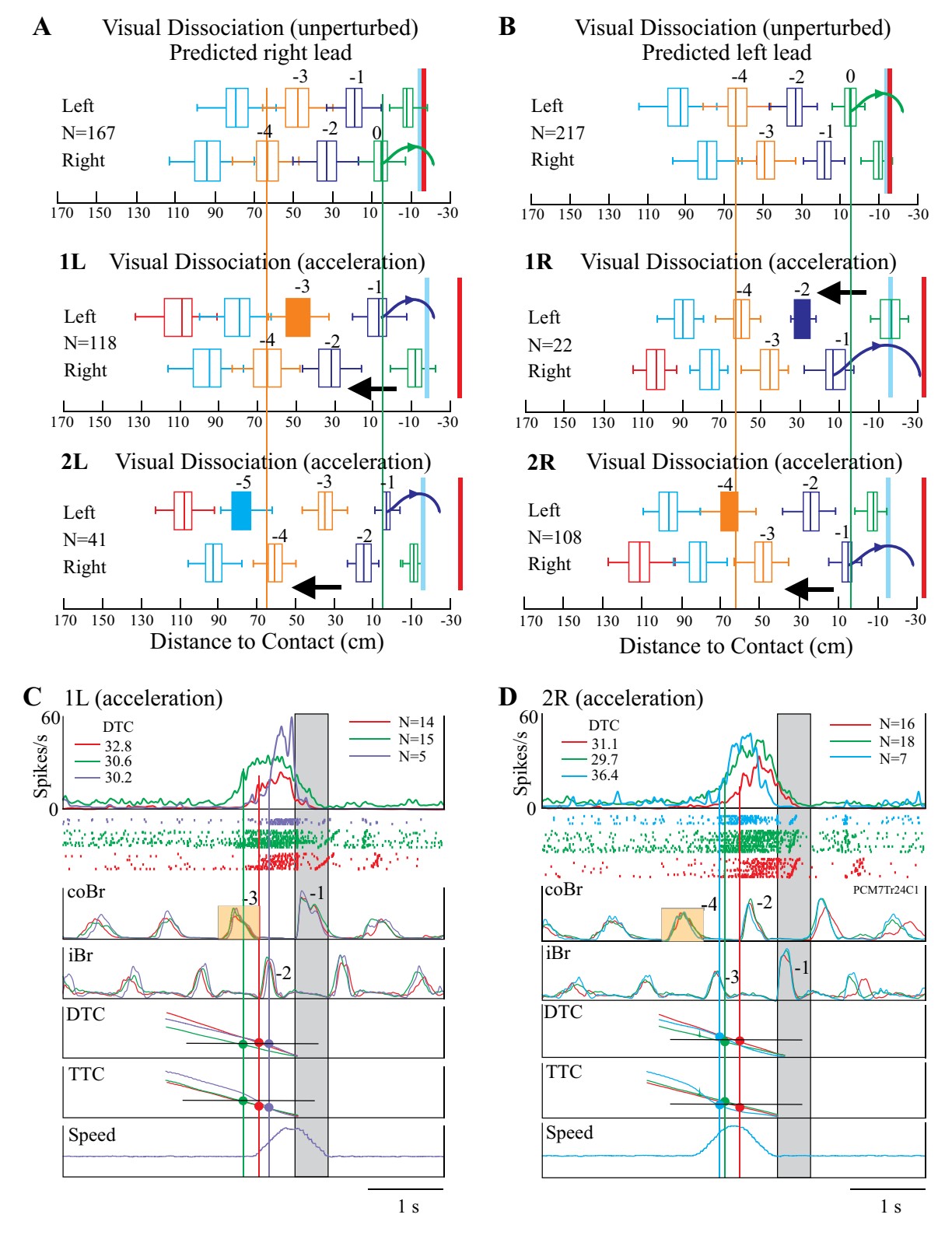

**Figure 7.** Effect of obstacle acceleration on behavior and cell discharge. **A and B** illustrate the distance of the cat from the obstacle at the onset of flexor muscle activity in the left and right limb lead condition (same representation as in **Figure 1**). The filled box indicates the step in which we applied the acceleration. The steps are color-coded to identify each step before the step over the obstacle together with the pairs of steps, organized in a right/left manner in A and left/right in B. The red vertical line indicates the approximate expected location of the obstacle in the absence of

*Figure 7 continued on next page*

*Figure 7 continued*

acceleration. The blue vertical line indicates the approximate position of the obstacle at the end of the steps reverses during the acceleration. Details are in the text. (C, D) organized as for *Figures 3–4*, with the bottom trace indicating the speed of the obstacle; note we applied the acceleration 200 ms after the onset of activity in the coBr. Data are shown for the matched (red), visual dissociation (green), and acceleration (purple, 1L or cyan, 2R) task. In C, accelerations correspond to those illustrated in the middle trace of A (1L condition) while in D they correspond to those illustrated in the bottom trace of B (2R condition). Small orange rectangles indicate the coBr burst used to trigger the acceleration and correspond to the colored boxes in A, B. Similarly, numbers beside the Br bursts correspond to the numbers identifying steps in A, B. Same cell as illustrated in *Figure 3*. (*Figure 7—source data 1* and *2*). *Figure 7—figure supplement 1*: Effect of acceleration on gait pattern. *Figure 7—figure supplement 2*: Relationship between time and distance for the visual dissociation and the 1L acceleration task. *Figure 7—figure supplement 3*: Relationship between time and distance for the visual dissociation and the 2R acceleration task.

DOI: https://doi.org/10.7554/eLife.28143.022

The following source data, source code and figure supplements are available for figure 7:

**Source data 1.** Source data for box plots in *Figure 7A*.
DOI: https://doi.org/10.7554/eLife.28143.026
**Source data 2.** Source data for box plots in *Figure 7B*.
DOI: https://doi.org/10.7554/eLife.28143.027
**Source code 1.** Script for data in *Figure 7*.
DOI: https://doi.org/10.7554/eLife.28143.028
**Figure supplement 1.** Effect of acceleration on gait pattern.
DOI: https://doi.org/10.7554/eLife.28143.023
**Figure supplement 2.** Relationship between time and distance for the visual dissociation and the 1L acceleration task.
DOI: https://doi.org/10.7554/eLife.28143.024
**Figure supplement 3.** Relationship between time and distance for the visual dissociation and the 2R acceleration task.
DOI: https://doi.org/10.7554/eLife.28143.025

The constant relationship with DTC for this neuron held also for the 2R acceleration condition as illustrated in *Figure 7D*. In this condition, cell discharge in the visual dissociation condition began 723 ms before the step over the obstacle at a DTC of 29.7 cm. We applied the acceleration in step −4, when the obstacle was still 56 cm from the cat and, as a result, obstacle velocity had almost returned to its pre-acceleration speed when the cell began to discharge (cyan trace) at a DTC of 36.4 cm and 733 ms before the step over the obstacle (see *Figure 7—figure supplement 3*). Therefore, an acceleration occurring prior to the predicted time of onset and the predicted DTC did not modify the onset of the cell discharge.

Most cells displayed similar changes in activity to those illustrated in *Figure 7* in response to acceleration of the obstacle. *Figure 8A,B* illustrate two other DTC-related cells in which acceleration modified the onset and the slope of the onset of the cell discharge. An acceleration just before the step over the obstacle (1L condition, purple traces in *Figure 8A,B*) produced similar changes to those observed in *Figure 7C*, in that both cells showed a relatively later onset during the acceleration than during the unperturbed visual dissociation task. In both cells, the DTC at which the cell discharged during the acceleration was similar to that obtained in the matched and the visual dissociation tasks (upper left of *Figure 8A,B*).

A similar, constant, relationship for a TTC-related cell is illustrated in *Figure 8C*. In this example, TTC remained almost constant for the matched and visual dissociation tasks, as well as during the 1L and the 2L acceleration conditions.

The onset of cell discharge during the 1L condition of the acceleration task was delayed with respect to onset during the visual dissociation task for the vast majority (33/36) of cells tested in this condition (*Figure 8D*). We found a significant delay in 19/36 of these cells (t-tests, $p<0.05$). Cell onset was also relatively delayed for most (8/9) cells following acceleration in the 1R condition and was significant in 2/9 cells (*Figure 8D*). However, the onset of discharge activity was less frequently delayed in the 2L and the 2R conditions, in which we found significant differences in only 1/9 and 2/26 cells, respectively (*Figure 8E*). In general, acceleration was less likely to modify the onset of cell discharge the earlier we applied it.

To determine whether all cells maintained the same constant relationship with distance or time in trials with accelerations, we repeated the ANOVA analysis described for *Figures 4–5*, with the addition of each type of acceleration in turn to the calculation. We found that in the majority of cases,

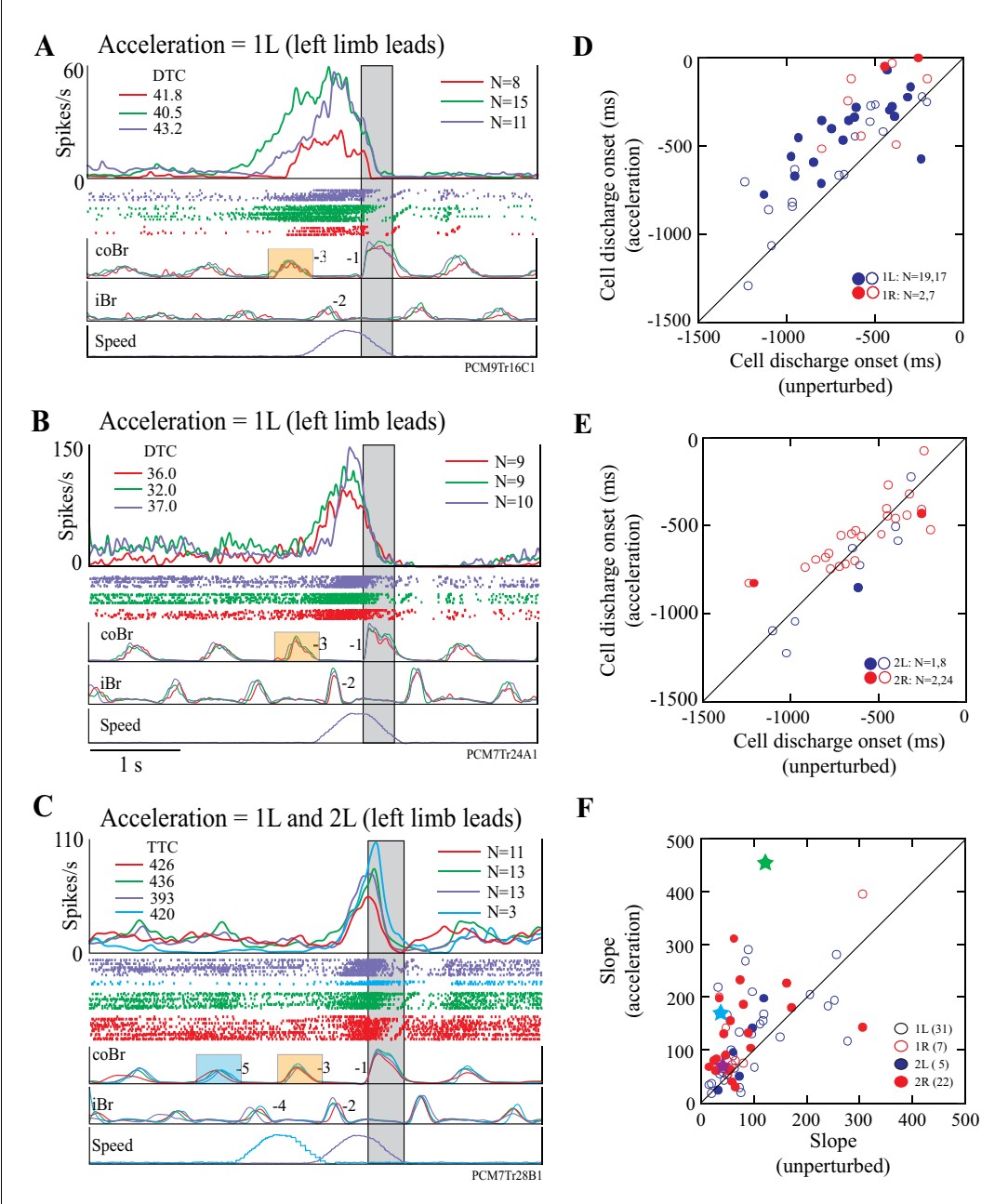

**Figure 8.** Examples and summary of effects of acceleration. (A,B) Example discharge of two DTC-related cells to acceleration of the obstacle in the 1L condition. (C) Example of a TTC-related cell to acceleration in the 1L and the 2L conditions. Figures organized as for *Figure 7C*. (D,E) Onset of cell discharge during accelerations as a function of the onset during the unperturbed visual dissociation task. Values are relative to the onset of the coBr during the step over the obstacle. (D), accelerations 1 step before (1L, 1R); (E), 2 steps before (2L, 2R). Filled circles in (D,E) indicate cells showing a significant change in the two conditions as determined by a t-test (p<0.05); open circles indicate non-significant values. (F) Slope of the cell discharge during the acceleration as a function of the slope during the unperturbed visual dissociation task. Slopes from all four acceleration conditions are illustrated (see key). Stars indicate the examples illustrated in *Figures 7C* and *8A,B* (see text). Slopes were calculated from averaged traces as (max discharge– discharge at cell onset)/Δt. Boxes on the coBr trace in A-C indicate the burst used to trigger the acceleration. (*Figure 8—source data 1* and *2*).

DOI: https://doi.org/10.7554/eLife.28143.029

The following source data and soruce codes are available for figure 8:

**Source data 1.** Source data for graphs in *Figure 8D,E*.
DOI: https://doi.org/10.7554/eLife.28143.030
**Source data 2.** Source data for graph in *Figure 8F*.

*Figure 8 continued*

DOI: https://doi.org/10.7554/eLife.28143.031

**Source code 1.** Script for data in *Figure 8*.

DOI: https://doi.org/10.7554/eLife.28143.032

the onset of cell discharge maintained a constant relationship to either distance (e.g. *Figure 5A*) or time (e.g. *Figure 5B*). We could test 20 acceleration conditions for 8/14 cells showing a relationship to DTC, and in most of these (18/20), the relationship with DTC was maintained during the different acceleration conditions (7/9, 1L; 2/2, 1R; 3/3, 2L; 6/6, 2R). Similarly, we tested 25 acceleration conditions for 8/15 cells with a constant relationship to TTC and most (16/25) equally maintained this relationship with the accelerations (6/11, 1L; 5/5, 1R; 1/1, 2L; 4/8, 2R).

Importantly, most cells showed a marked increase in the slope of the increase in discharge frequency during the acceleration, particularly during the 1L condition (*Figure 8F*). For example, in the example illustrated in *Figure 7C*, the slope increased from a value of 35.5 spk.s$^{-2}$ in the unperturbed visual dissociation task to 170.7 spk.s$^{-2}$ during the acceleration (cyan symbol in *Figure 8F*), while in *Figure 7D* the increase went from 46.2 sps.s$^{-2}$ to 90.3 spk.s$^{-2}$. Similarly, we found a clear increase in slope for the examples illustrated in *Figure 8A,B* (purple and green symbols, respectively). Altogether, three quarters of our examples (49/67, 73%) showed an increase in the slope of the activity during the acceleration. We observed the largest changes in slope in the 1L and 2R conditions.

The increase in the slope of the cell discharge for the 1L and the 2R condition is well illustrated in the population averages of *Figure 9A,B*, respectively. These population plots show that the slope of the discharge during the acceleration was 3.3 times greater than during the unperturbed visual dissociation task for the 1L condition and 2.9 times greater in the 2R condition. An increased slope is also very clear for the 1R condition in which the late onset of the acceleration initiated a rapid change in the limb sequence (see *Figure 9—figure supplement 1A*) that we consider more of an online correction than a planned gait modification. However, we did not observe any noticeable change in the slope of the discharge for the 2L condition (see *Figure 9—figure supplement 1B*), in which the acceleration occurred earlier (average of 1608 ms) than the onset of discharge activity in most cells. It is probable that the increased slope of the discharge in the conditions in which the

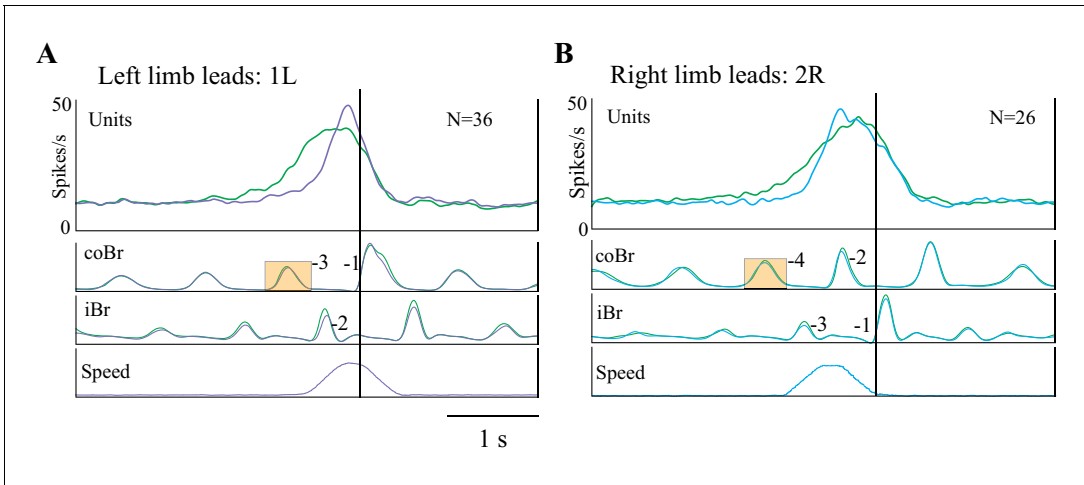

**Figure 9.** Population activity during obstacle acceleration. (**A,B**) Population discharge activity is shown for all cells for which we applied the 1L and 2R protocols. Green traces show averaged activity in the visual dissociation task; the purple trace indicates the 1L acceleration condition; and the cyan trace indicates 2R condition. Data aligned to the onset of activity in the Br during the step over the obstacle. Boxes on the coBr trace in A,B indicate the burst used to trigger the acceleration (*Figure 9—figure supplement 1*: Population averages during Acceleration Task).

DOI: https://doi.org/10.7554/eLife.28143.033

The following figure supplement is available for figure 9:

**Figure supplement 1.** Population averages during acceleration task.

DOI: https://doi.org/10.7554/eLife.28143.034

acceleration relatively delayed cell onset provides information on the rate of change of gap closure (see Discussion).

## Discussion

In this manuscript, we demonstrate two distinct neuronal populations in the PPC (primarily in area 5b) whose properties support a role in signaling the relationship between body position and object location during walking. One population's discharge activity increased in relation to particular distances-to-contact with an obstacle. The second population's discharge activity increased in relation to particular times-to-contact with an obstacle. We propose that walking animals use this information to appropriately modify the spatial and temporal parameters of the gait modification required to negotiate a moving obstacle. These results emphasize a contribution of populations of neurons in the PPC to the control of locomotion that goes beyond the control of limb-specific activity related to limb trajectory or the EMG patterns required to execute the step over the obstacle. Instead, we suggest that this pattern of activity is intricately implicated in the transformation of information obtained from vision into an appropriate motor plan that can be used for obstacle avoidance.

The presence of cells discharging to an advancing object is compatible with the existence of cells in multiple areas, including the PPC, that respond to optic flow stimuli (see Introduction). However, the cells in our study did not discharge purely to the visual stimulus, in which case they might have been expected to discharge throughout the period that the obstacle was visible (10–12 steps before the step over the obstacle). Rather, they began to discharge only when the obstacle was at a fixed DTC or TTC from the cat. Moreover, most of the cells maintained this relationship even when we accelerated the obstacle toward the cat. This suggests that these cells are tuned to respond to objects only when they are within a limited range of DTC or TTC. A similar tuning of cell responses to distance in response to looming stimuli is found in the ventral intraparietal area (VIP) of the PPC in non-human primates (*Colby et al., 1993*; *Graziano and Cooke, 2006*; see also *Hadjidimitrakis et al., 2015*), as well as in the premotor cortex (PMC, *Graziano et al., 1997*). Graziano (*Graziano and Cooke, 2006*) has proposed that such cells may play a role in defensive and avoidance behavior.

A similar function might be ascribed to the responses recorded in our task in which the cat must interact with the advancing obstacle by modifying its gait pattern to step over it. The discharge activity should therefore not be considered simply in terms of the visual stimulus but rather in the context of a coordinated and planned motor activity. In this respect, cells responding at fixed DTC and TTC might be considered as providing important context-dependent information that is used to plan the upcoming gait modification. Moreover, as in the experiments referenced above with respect to the VIP and PMC, the cells in our study discharged in a staggered manner over a limited range of DTC and TTC. We believe that this feature of the discharge activity would provide a means for animals to continually monitor the rate of gap closure over the range in which a gait modification needs to be planned. The ability to continually monitor obstacle location over time would facilitate the detection of any non-linearities in the rate of gap closure and might be particularly important in helping the cats negotiate the obstacle when it is accelerated. In a more natural situation, the sequential activation of both DTC and TTC-related cells would be necessary for estimating the gap with a prey moving at unpredictable speeds, different from those of the predator.

It is important to emphasize that although the cells discharged at fixed DTC/TTC before the gait modification, cell discharge continued until the step over the obstacle. As such, we believe that the increased discharge prior to the step over the obstacle is not a pure visual response but rather represents the starting point of a complex sensorimotor transformation involved in planning the gait modification. In this view, visual input is essential for initiating the sensorimotor transformation, but once initiated planning can continue in the absence of continual visual input.

The results of our earlier lesion studies also support a role for the PPC cells in the sensorimotor transformation required to step over the obstacle. Lesion of the PPC region in which we recorded these cells results in a marked locomotor deficit defined by an inability to appropriately place the plant limb in front of the obstacle (*Lajoie and Drew, 2007*). We have previously discussed the reasons that we believe that this deficit is indicative of an error in planning rather than one of perception or action. We propose that in the absence of information about the relative location of the

obstacle and of the rate of gap closure provided by the cells in the PPC, the cat is unable to determine where to position its leg in front of the obstacle and when to start the gait modification.

The ensemble activity of the population of cells demonstrates a progressive increase in discharge rate up to the time of the gait modification. A similar ramp increase in cell discharge as TTC progressively decreases has been observed in the motor cortex in monkeys trained to intercept a simulated object with their arm (*Merchant and Georgopoulos, 2006*). In our task, we propose that this ramp increase in the population discharge provides a signal that indicates the imminent requirement to make the gait modification. This ramp discharge is reminiscent of that observed in several structures and in many tasks in which motor activity is self-initiated (*Lebedev et al., 2008*; *Maimon and Assad, 2006a*, *2006b*; *Merchant and Georgopoulos, 2006*) and is particularly prevalent in tasks in which a decision to move on the basis of ambiguous or delayed information is required (*Cisek and Kalaska, 2005*; *Cisek and Kalaska, 2010*; *Roitman and Shadlen, 2002*; *Thura and Cisek, 2014*). *Thura and Cisek (2014)* refer to the time at which such a decision is made as the time of commitment. We consider that the peak of activity in our population of cells also indicates a time of commitment, at which point the cat initiates the gait modification. In this respect, it is pertinent that when we accelerated the obstacle, we found a greater slope of the activity between the onset of cell discharge and the onset of the gait modification than in the absence of acceleration. This increased slope allowed the cells to reach a similar peak value in the shorter time available to the cat to make the gait modification.

Although we believe that the activity that we observed in this task forms part of the sensorimotor transformation that leads to the gait modification, its function has to be discussed in light of the fact that the discharge is limb-independent, that is, it is identical regardless of whether the left or the right limb is the first to step over the obstacle. This is contrary to one study (*Bernier et al., 2012*) that suggests that activity in the PPC is not expressed until the effector limb has been specified, and then only for the contralateral limb (although see *Chang et al., 2008*) While this disparity could relate to species or area-specific differences, we believe that the nature of the task requirements is probably the main determinant. In the experiments of *Bernier et al. (2012)*, subjects were static and an external cue specified the arm to move. In our task, the cat is walking and the leg that will step over the obstacle is not pre-defined. The cat must process the incoming information on gap closure and must integrate the planned gait modification into its natural rhythm. As such, we suggest that the decision as to which limb to use to step over the obstacle should be viewed as an emergent property of the task rather than as an instructed movement or a decision made in advance of the movement as in the tasks referenced above.

One possible manner in which information on gap closure could be used to initiate a gait modification is illustrated in *Figure 10*. In this conceptual model, we presume that there is an integration of the gap closure signal with a second signal that provides information on the state of the limb. The integration with limb state ensures that the gait modification will only be initiated when the limb is in the appropriate state, that is, at the end of the stance phase and ready to initiate the transfer of the limb over the obstacle. We propose that this integration proceeds bilaterally and the limb selected to be the first to step over the obstacle depends on which side wins the competition. Although we do not wish to unduly speculate on where this integration would occur, we would note that all of the cells recorded in this study were located in layer V and therefore project to subcortical structures, including the basal ganglia and the cerebellum.

In conclusion, our results provide new insights, at the single cell level, into the sensorimotor transformations that underlie the control of visually guided walking. The demonstration of populations of cells that can serve to provide information on gap closure and potentially initiate precise gait changes is a novel contribution to our understanding of the control mechanisms used to guide locomotion. Taken together with the results from studies in various species that show a contribution of the PPC to spatial navigation (*Calton and Taube, 2009*; *Harvey et al., 2012*; *Whitlock, 2014*), it is possible that the PPC may have a privileged position in contributing to our ability to plan the gait adjustments needed to negotiate a complex environment.

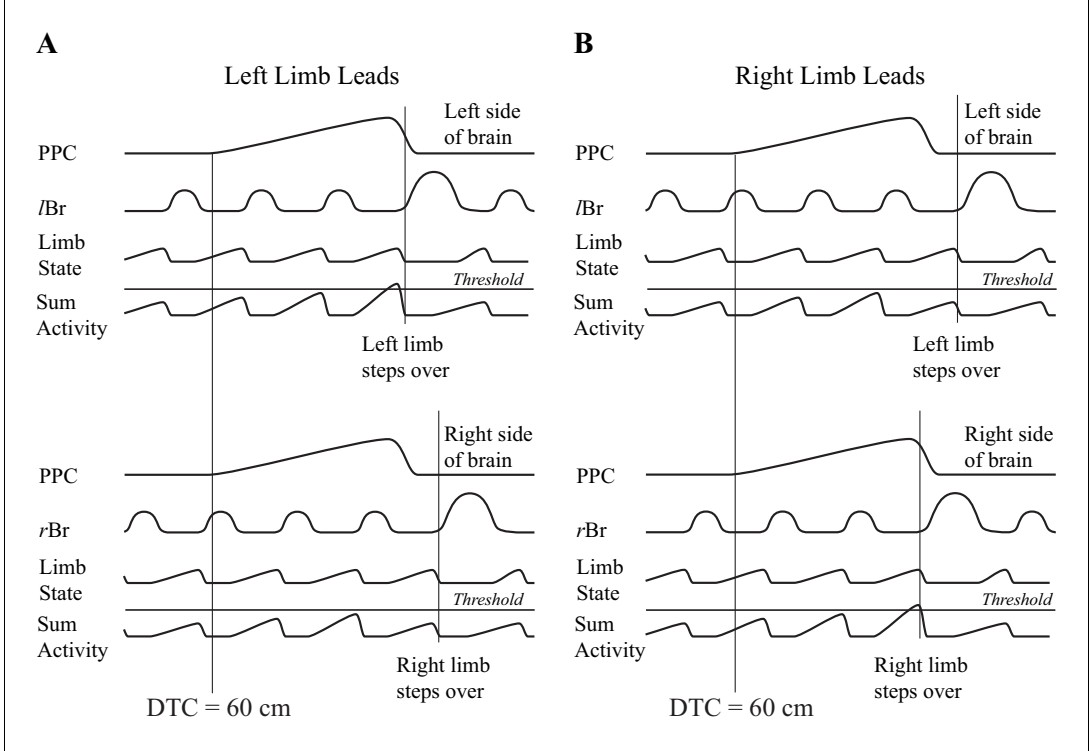

**Figure 10.** Conceptual model illustrating a possible mechanism for limb selection. Left (**A**) and right (**B**) limb lead conditions. The top parts of A,B show activity on the left side of the brain and the bottom parts of A,B, show activity on the right side of the brain. In each part of the figure, the traces represent, from top to bottom, the population neuronal activity in the PPC, EMG activity from the left or right Br, a signal representing limb state, and the integrated (summed) activity of these two traces. We suggest that the signal from the PPC, providing information on gap closure, is integrated with a second signal providing information on limb state, on each side of the brain. Whichever integrated signal crosses a threshold level first determines which limb will be selected to be the first to negotiate the obstacle. The population neuronal activity in this illustration is assumed to start 60 cm before the step over the obstacle in both the left and right limb leads conditions. Whether the animal steps over with the left or right limb depends on when cell activity begins with respect to the ongoing locomotor activity of the animal. In A, a threshold level is achieved first on the left side (top illustration), whereas in B, the threshold level is achieved first on the right side (bottom illustration). Note that the integrated value will only cross the threshold level when the signal from the moving limb shows it to be in a state that is appropriate to negotiate the obstacle (e.g. the end of stance).
DOI: https://doi.org/10.7554/eLife.28143.035

## Materials and methods

### Training and task

We performed experiments on the same two cats (PCM7 and PCM9) that we previously used in other experiments (*Marigold and Drew, 2011*). We initially trained the cats to walk on a treadmill at a constant speed of 0.45 m.s$^{-1}$ (unobstructed locomotion) and then trained them to step over obstacles attached to a second belt that moved at the same speed as the treadmill (matched task). Subsequently, we trained each cat to step over the obstacles that were advanced at a slower speed (0.3 m.s$^{-1}$) than the treadmill belt on which the cat walked (visual dissociation task: *Drew et al., 2008*; *Lajoie and Drew, 2007*). Finally, we habituated the cats to a third task in which the obstacle accelerated toward them several steps prior to the step over the obstacle (acceleration task). Accelerations consisted of a ramp increase from 0.3 m.s$^{-1}$ to ~0.65 m.s$^{-1}$ over a period of 450 ms and a symmetrical decrease back to baseline levels. Note that cats very rarely, if ever, hit the obstacle, even in conditions in which the acceleration occurred just before the planned gait modification.

Two obstacles (cylindrical in shape and each of 10 cm cross-section), separated by 3 m, were attached to the treadmill belt. Depending on the speed of the treadmill, the cat generally made 12–14 steps (6–7 step cycles) between each obstacle. The obstacle became visible to the cat ~2 m before the step over the obstacle. The obstacle was therefore visible to the cat for 5–7 s before the gait modification, during which time the cat made 10–12 steps.

All handling and surgical procedures followed the recommendations of the Canadian Council for the Protection of Animals, and the animal ethics committee at the Université de Montréal approved the experimental protocols (#12_082).

## Surgery

Once trained, we prepared the cats for surgery in aseptic conditions as described in previous papers (*Andujar et al., 2010*; *Drew, 1993*; *Marigold and Drew, 2011*). In brief, based on an MRI taken 1–2 weeks before the surgery, we placed a stainless steel baseplate (internal dimensions = 12 by 6 mm) over the right PPC and then formed a recording chamber around it with dental acrylic. We implanted pairs of Teflon-insulated, stainless steel braided wires into selected fore- and hindlimb muscles to record electromyographic (EMG) activity. The wires ran subcutaneously to a connector on the cranium. To allow for antidromic identification of projection neurons in layer V of the PPC, we inserted microwires into the cerebral peduncle by using a harpoon assembly (*Drew, 1993*; *Palmer, 1978*). Following the surgery, we administered buprenorphine (5 µg/kg) for a period of 72 hr, and antibiotics for a period of 10 days. Experiments started 1 week after the surgery.

## Protocol

In each recording session, we introduced a conventional glass-insulated tungsten microelectrode (0.5–1.5 MΩ) into the PPC using a custom-made microdrive. We advanced the electrode until stimulation of the electrodes in the cerebral peduncle produced antidromic discharges either in an isolated unit or in smaller units in the background. This provided evidence that the electrode had reached layer V. We recorded cell activity from well-isolated single units while the cat walked on the treadmill in the matched task until approximately 10 steps over the obstacle with each leg as the lead limb had been made. After slowing the obstacle (visual dissociation task), locomotion continued until we collected approximately five steps with each leg. In selected subsequent steps, we accelerated the obstacle toward the cat several steps before the step over the obstacle. In these steps, acceleration always occurred 200 ms after the onset of activity in the left brachialis (Br) or cleidobrachialis (ClB) contralateral to the recording site. Steps in which the obstacle accelerated were interspersed unpredictably with steps where the obstacle continued at its pre-set speed. The entire data collection period occupied 15–20 min, although we sometimes lost cell stability before we could complete the recording session.

We band-pass filtered EMG activity at 100–475 Hz and digitized it online at a frequency of 1 KHz. To discriminate cells offline, we digitized cell activity at a frequency of 100 KHz. In all experiments, a six-camera Vicon motion analysis system recorded, at 100 Hz, the position of light-reflecting points placed on a rod attached to the head of the cat and along the length of each obstacle. We synchronized cell, EMG, and motion data for later analyses.

## Analysis

We discriminated single units offline on the basis of waveform amplitude and shape. For sections of data with stable action potentials and with stable locomotion, we marked the onset and offset of activity in the left, contralateral (co) and right, ipsilateral (i) Br or ClB EMG for every step during the entire locomotor sequence. This allowed us to identify each step as a step over the obstacle, one or more steps before the obstacle, or the step after the obstacle. We further identified steps as to whether the left, contralateral forelimb (left limb leads condition) or the right, ipsilateral forelimb (right limb leads condition) stepped over the obstacle first — in previous publications from this laboratory, these are referred to as lead and trail conditions, respectively. Cell activity during unobstructed locomotion was calculated on the basis of the discharge activity five steps before the step over the obstacle, combining all tasks. As such, activity during unobstructed locomotion is interspersed with the data obtained during the steps over the obstacle and was obtained from the entire recording period.

To determine the temporal relationships between cell discharge activity and different behavioral events on a trial-by-trial basis, we transformed cell discharge for each trial into an instantaneous frequency (1000/interspike interval) and filtered it at 50 Hz (fourth order Butterworth algorithm). By using interactive software, we calculated the onset of cell activity relative to the step over the obstacle for each trial as an increase or decrease of activity that exceeded 2SD of the cell discharge that

occurred between 2.5 and 3 s before the step over the obstacle (see *Figure 3—figure supplement 1*). We then calculated the distance of the obstacle from the cat (distance-to-contact; DTC) and the time-to-contact (TTC) with the obstacle at the moment of the onset of change in cell discharge, for each individual trial, by measuring the relative distance of the rod on the cat's head from the obstacle. To make box plots of the time of cell onset for each task (matched, visual dissociation, and acceleration) and condition (left and right limb lead), we calculated median and interquartile ranges (IQR). We removed data values that exceeded 1.5 * IQR from the analysis. On average, this removed 2.5% of the trials (we recorded an average of 62 trials/cell).

After removal of outliers, we calculated average discharge rates of cell activity (bin width = 2 ms) by synchronizing activity to the onset of the Br or ClB. We used these average displays to determine if cells were step-related or step-advanced and whether they were limb-independent or limb-dependent (*Andujar et al., 2010*). A change in discharge activity beginning >200 ms before the onset of activity in the Br or ClB differentiated step-advanced cells from step-related cells. We defined a limb-independent cell as one in which discharge activity (as determined from the averaged displays) ended at approximately the same time (<200 ms difference) with respect to the coBr (coClB) during the left, contralateral forelimb, lead condition and with respect to the iBr (iClB) during the right, ipsilateral forelimb, lead condition (see *Andujar et al., 2010*; *Marigold and Drew, 2011*).

We used Systat V13 (Systat Software Inc.) for all statistical analyses. One-way ANOVAs determined the effect of distance and time on the time of onset of cell discharge (significant effects determined at the alpha = 0.01 level). Significant differences between pairs of values using t-tests were determined using an alpha of 0.05. When multiple comparisons were made we used a Bonferroni correction. To create an index of the difference in discharge with respect to DTC and TTC, we calculated the difference between the averaged standardized discharge rate (Z score) during the visual dissociation task (left + right/2) and that during the matched task (left +right/2). In this index, cells that showed constant activity in both tasks would have TTC and DTC indexes close to zero. Those cells that show a constant relationship to DTC will have a DTC index centered around 0.0 and a TTC index close to 1.0, while cells with a constant relationship to TTC will have a TTC index centred around 0 and a DTC index close to −1.0. To determine the extent to which this index succeeded in identifying the two categories of cell, we also performed a bootstrapping exercise in which we used a replacement protocol to create new datasets for each cell (see legend to *Figure 5—figure supplement 1*). Calculations were performed 1000 times for each cell.

We included cells in the analysis on the following bases: (1) The cells were located within the caudal bank of the ansate sulcus or the adjoining caudal gyrus, corresponding to area 5b and the border with area 7. (2) The cells were located within cortical layer V, as determined on the basis of the antidromic stimulation (43/67 cells were antidromically activated, as determined by collision with spontaneous action potentials, and the other 24/67 cells were adjacent to identified cells). (3) All cells discharged >200 ms before the step over the obstacle (step-advanced cells, *Andujar et al., 2010*). (4) All cells manifested a limb-independent pattern of activity, allowing us to compare activity when the left and right limbs stepped over the obstacle. (5) We only included cells in the analysis if we recorded at least five steps over the obstacle during unperturbed walking and three steps for each condition during the acceleration task.

## Histology

At the end of the series of experiments, we made small lesions (30–50 µA) in selected locations within the recording chamber and perfused the cat *per cardia* with formalin. We sectioned the brain in the sagittal plane (40 µm sections) and stained it with cresyl violet. The depth of layer V (as determined during the recordings) and the terminal lesions were used to determine the location of the electrode penetrations.

## Acknowledgements

We would like to thank M Bourdeau, N De Sylva, P Drapeau, C Gauthier, F Lebel and J Soucy for technical assistance in the performance and analysis of these experiments. Dr. Kim Lajoie also participated in some of these experiments. We thank Drs. Elaine Chapman, Paul Cisek and John Kalaska for helpful comments on this manuscript. This work was supported by an operating grant (MOP-

53339) from the CIHR and an infrastructure grant from the FRSQ. DS Marigold received salary support from the CIHR.

## Additional information

### Funding

| Funder | Grant reference number | Author |
| --- | --- | --- |
| Canadian Institutes of Health Research | MOP-53339 | Trevor Drew |
| Fonds de Recherche du Québec - Santé | Infrastructure Group Grant | Trevor Drew |

The funders had no role in study design, data collection and interpretation, or the decision to submit the work for publication.

### Author contributions
Daniel S Marigold, Formal analysis, Writing—original draft, Writing—review and editing; Trevor Drew, Conceptualization, Formal analysis, Writing—review and editing

### Author ORCIDs
Daniel S Marigold (iD) http://orcid.org/0000-0001-6610-9058
Trevor Drew (iD) http://orcid.org/0000-0001-5199-3581

### Ethics
Animal experimentation: All handling and surgical procedures followed the recommendations of the Canadian Council for the Protection of Animals, and the animal ethics committee at the Université de Montréal approved the experimental protocols.

### Decision letter and Author response
Decision letter https://doi.org/10.7554/eLife.28143.037
Author response https://doi.org/10.7554/eLife.28143.038

## Additional files
### Supplementary files
• Transparent reporting form
DOI: https://doi.org/10.7554/eLife.28143.036

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
