## [Decision Letter]

Thank you for submitting your article "Posterior parietal cortex estimates the relationship between object and body location during locomotion" for consideration by *eLife*. Your article has been favorably evaluated by David Van Essen (Senior Editor) and two reviewers, one of whom, Jody C Culham (Reviewer #1), is a member of our Board of Reviewing Editors. The following individual involved in review of your submission has agreed to reveal their identity: Stephen Lomber and Carmen Wong (Reviewer #2).

The reviewers have discussed the reviews with one another and the Reviewing Editor has drafted this decision to help you prepare a revised submission.

Summary:

The reviewers thought the manuscript makes a valuable contribution to the literature. Reviewer #1 stated, "This paper makes an important contribution to revealing the role of PPC in visually guided locomotion, which is far less studied than other functions such as reaching and grasping. The design is elegant, the results straightforward, and the paper well written." Reviewer #2 stated, "The authors employ elegant experimental manipulations to examine complex locomotor control at the single unit level. This manuscript describes very compelling results and it was a pleasure to read. The manuscript is well written and detailed, and the data analysis is extremely comprehensive. The figures are very well thought-out and relatively straight-forward to understand. Summary Figure 8 is particularly well done."

Recommended revisions:

Neither of the reviewers made comments that are essential to address in the revision (beyond outright errors). However, we recommend that you seriously consider a number of suggestions intended to improve the manuscript.

1) Reviewer #1/RE would like you to clarify which aspects of the data are necessitated by the criteria used to categorize cells.

Specifically, she states:

I only have one substantive comment and will leave it up to the authors if/how they want to address it. As a neuroimager (and outsider to neurophysiology), I think neurophysiology research would benefit from deeper consideration of some of the statistical issues that have arisen in neuroimaging with respect to the "non-independence problem", also called "circular analysis" or "double dipping" (see esp. Kriegeskorte, 2009, Nature Neuroscience). The present paper doesn't suffer from the most egregious form of this that occurs in neurophys where even random data would show the claimed effects (e.g., classifying cells based on preferred stimuli and then showing PSTHs illustrating that – surprise! – they have a preference for the preferred stimuli). Nevertheless, I did find myself wondering (1) how much of the data presented must necessarily be true based on the criteria used to classify the cells as DTC or TTC (especially in Figure 4); and (2) how reliable the classifications of a cell would be if evaluations were done on an independent data set (e.g., classing on even trials, testing on odd)?

Concerns about non-independence turned into a hysterical witch hunt in neuroimaging until saner minds prevailed (Poldrack et al., 2009, SCAN), pointing out that much of the problem can be avoided by simply acknowledging which aspects of the data are circular, possibly with exaggerated effect sizes and which are not. This should be done here. I realize the analyses here are standard for neurophysiology and, as per *eLife* policy, won't insist on new analyses (like split-half reproducibility) that are not essential to conclusions. However, I would like to nudge the authors (and indeed the neurophysiology community more broadly) to think about these issues more deeply and be clearer about circularity if present and promote the robustness of the results if not.

2) Both reviewers think that the manuscript would benefit from a more precise definition of PPC as area 5 and its border with area 7, especially in the Abstract. As is, the continued reference to PPC (in general) could leave the impression that these effects are found throughout when they may in fact be limited to area 5. It would also be beneficial to include a photo or diagram of the cat brain to show the location of the recording sites to readers that might not be familiar with brain motor cortex.

3) For the 67 cells included in this study, specify how many were recorded from each animal and the relative numbers of TTC and DTC cells studied.

4) Subsection “Cell discharge during matched and visual dissociation tasks”, first paragraph: As the average onset of cell discharge is stated for each condition, the corresponding distances could be included to demonstrate the similarity in their value, and emphasize that this example cell is a DTC cell.

5) Results reported from the obstacle acceleration condition nicely detail example DTC cells, but do not show any example TTC cells. Additional panels in Figure 7 or a similar figure to Figure 7 depicting example TTC cells recorded during the obstacle acceleration condition would be useful.

6) While performing the task, did either animal ever hit the obstacle? It was mentioned in the last paragraph of the Results section that steps could have been corrected online. Was such correction preceded by contact with the obstacle?

---

## [Author Response]

Recommended revisions:Neither of the reviewers made comments that are essential to address in the revision (beyond outright errors). However, we recommend that you seriously consider a number of suggestions intended to improve the manuscript.1) Reviewer #1/RE would like you to clarify which aspects of the data are necessitated by the criteria used to categorize cells.Specifically, she states:I only have one substantive comment and will leave it up to the authors if/how they want to address it. As a neuroimager (and outsider to neurophysiology), I think neurophysiology research would benefit from deeper consideration of some of the statistical issues that have arisen in neuroimaging with respect to the "non-independence problem", also called "circular analysis" or "double dipping" (see esp. Kriegeskorte, 2009, Nature Neuroscience). The present paper doesn't suffer from the most egregious form of this that occurs in neurophys where even random data would show the claimed effects (e.g., classifying cells based on preferred stimuli and then showing PSTHs illustrating that – surprise! – they have a preference for the preferred stimuli). Nevertheless, I did find myself wondering (1) how much of the data presented must necessarily be true based on the criteria used to classify the cells as DTC or TTC (especially in Figure 4); and (2) how reliable the classifications of a cell would be if evaluations were done on an independent data set (e.g., classing on even trials, testing on odd)?Concerns about non-independence turned into a hysterical witch hunt in neuroimaging until saner minds prevailed (Poldrack et al., 2009, SCAN), pointing out that much of the problem can be avoided by simply acknowledging which aspects of the data are circular, possibly with exaggerated effect sizes and which are not. This should be done here. I realize the analyses here are standard for neurophysiology and, as per eLife policy, won't insist on new analyses (like split-half reproducibility) that are not essential to conclusions. However, I would like to nudge the authors (and indeed the neurophysiology community more broadly) to think about these issues more deeply and be clearer about circularity if present and promote the robustness of the results if not.

As the reviewer states, we do not believe that we have introduced any selection bias. The only selection prior to analysis was that all included cells discharge prior to the step over the obstacle. This is stated in the Materials and methods and is a necessary condition for a cell involved in planning. Other than that condition, cells were free to be categorised as DTC- or TTC-related, or neither, without further selection. With respect to the classification of the cells, we did not feel that we could classify and test on alternate trials because of the relatively small number of trials for each cell. Instead, we performed a bootstrapping exercise on the data used to calculate the index illustrated in Figure 4D (Now Figure 5D). We used a replacement protocol to calculate 1000 datasets for each of the DTC- and TTC-related cells in Figure 5D. The 95% ellipses illustrated in new Figure 5—figure supplement 1 support our contention that these cells form two categories. See the fifth paragraph of the subsection “Distinct populations of cells are related to DTC and TTC” and the fourth paragraph of the subsection “Analysis”.

2) Both reviewers think that the manuscript would benefit from a more precise definition of PPC as area 5 and its border with area 7, especially in the Abstract. As is, the continued reference to PPC (in general) could leave the impression that these effects are found throughout when they may in fact be limited to area 5. It would also be beneficial to include a photo or diagram of the cat brain to show the location of the recording sites to readers that might not be familiar with brain motor cortex.

We agree that we should have included more information on the recording sites. We have added a new figure (Figure 2) to the manuscript to address these comments. The figure includes a photograph of the brain of each cat used in the study, together with recording sites. Indeed, as the reviewers suggest, we recorded primarily from area 5b of the PPC and, most DTC- and TTC-related cells were, accordingly found in area 5b. We have modified the Abstract as well as added or modified text in several other places where we refer to our recording region (see subsection “Neuronal Database” and Discussion, first paragraph).

3) For the 67 cells included in this study, specify how many were recorded from each animal and the relative numbers of TTC and DTC cells studied.

We have added the pertinent information in two places: subsection “Neuronal Database” and subsection “Distinct populations of cells are related to DTC and TTC”, fourth paragraph.

4) Subsection “Cell discharge during matched and visual dissociation tasks”, first paragraph: As the average onset of cell discharge is stated for each condition, the corresponding distances could be included to demonstrate the similarity in their value, and emphasize that this example cell is a DTC cell.

As the next section specifically addresses the issue of distance-to-contact (and time-to-contact), we prefer to restrict the description to the time of discharge onset.

5) Results reported from the obstacle acceleration condition nicely detail example DTC cells, but do not show any example TTC cells. Additional panels in Figure 7 or a similar figure to Figure 7 depicting example TTC cells recorded during the obstacle acceleration condition would be useful.

We have modified Figure 7 (new Figure 8) and added an example of a TTC-related cell to this modified figure (Figure 8C) as requested. To avoid reducing the size of the illustrations we have removed the population analysis from Figure 8 and included them in a new figure (Figure 9).

6) While performing the task, did either animal ever hit the obstacle? It was mentioned in the last paragraph of the Results section that steps could have been corrected online. Was such correction preceded by contact with the obstacle?

The cats rarely, if ever, hit the obstacle, even in this very challenging condition. They are able to make very fast corrections of gait, as we previously remarked with respect to the responses to visual occlusion (Marigold and Drew, 2011). We have added a sentence to this effect in the Materials and methods (subsection “Training and task”, first paragraph).